# Predicting the Effect of miRNA on Gene Regulation to Foster Translational Multi-Omics Research—A Review on the Role of Super-Enhancers

**DOI:** 10.3390/ncrna10040045

**Published:** 2024-08-15

**Authors:** Sarmistha Das, Shesh N. Rai

**Affiliations:** 1Biostatistics and Informatics Shared Resource, University of Cincinnati College of Medicine, Cincinnati, OH 45267, USA; sarmistha.das@uc.edu; 2Cancer Data Science Center, University of Cincinnati College of Medicine, Cincinnati, OH 45267, USA; 3Division of Biostatistics and Bioinformatics, Department of Biostatistics, Health Informatics and Data Sciences, University of Cincinnati College of Medicine, Cincinnati, OH 45267, USA

**Keywords:** super-enhancer, miRNA, gene regulatory network, transcription factor, multi-omics

## Abstract

Gene regulation is crucial for cellular function and homeostasis. It involves diverse mechanisms controlling the production of specific gene products and contributing to tissue-specific variations in gene expression. The dysregulation of genes leads to disease, emphasizing the need to understand these mechanisms. Computational methods have jointly studied transcription factors (TFs), microRNA (miRNA), and messenger RNA (mRNA) to investigate gene regulatory networks. However, there remains a knowledge gap in comprehending gene regulatory networks. On the other hand, super-enhancers (SEs) have been implicated in miRNA biogenesis and function in recent experimental studies, in addition to their pivotal roles in cell identity and disease progression. However, statistical/computational methodologies harnessing the potential of SEs in deciphering gene regulation networks remain notably absent. However, to understand the effect of miRNA on mRNA, existing statistical/computational methods could be updated, or novel methods could be developed by accounting for SEs in the model. In this review, we categorize existing computational methods that utilize TF and miRNA data to understand gene regulatory networks into three broad areas and explore the challenges of integrating enhancers/SEs. The three areas include unraveling indirect regulatory networks, identifying network motifs, and enriching pathway identification by dissecting gene regulators. We hypothesize that addressing these challenges will enhance our understanding of gene regulation, aiding in the identification of therapeutic targets and disease biomarkers. We believe that constructing statistical/computational models that dissect the role of SEs in predicting the effect of miRNA on gene regulation is crucial for tackling these challenges.

## 1. Introduction

Gene regulation represents a collection of mechanisms by which cells increase or decrease the production of specific gene products. Cells from different tissues turn on specific genes and turn off others, thus contributing to tissue-related differences in the expression of the entire repertoire of genes. Aberrations in gene regulation may promote deviations from normal pathway activity, sometimes leading to complex diseases. The expression of any gene may be regulated due to DNA modification, transcriptional regulation, epigenetic changes, regulation by RNA, etc. While only a little over 1% genome code for protein is known [1], the remaining genome that was once thought of as junk has emerged as a crucial hub of gene regulators.

The gene regulatory elements, consisting of non-coding DNA sequences, serve as binding sites for transcription factors (TFs), which regulate (as activators/repressors) the transcription rate of genetic material from DNA to RNA. These regulatory mechanisms involve TF binding to DNA regions such as enhancers and promoters, which are often situated proximal to genes. Enhancers, characterized by multiple activator and repressor binding sites, play crucial roles in spatiotemporal gene expression and cellular memory through DNA looping [2] as well. Super-enhancers (SEs), are clusters of enhancers closely located in the genome. They exhibit high levels of transcriptional co-activators, active chromatin marks, and cell-type-specific TF motifs, distinguishing them from typical enhancers [3]. Notably, genetic mutations within enhancers have been linked to tumors and common diseases [4,5]. Numerous GWAS SNPs are notably enriched within SEs [6,7,8]. Chromatin immunoprecipitation followed by sequencing (ChIP-Seq) is a pivotal technology for mapping genome-wide protein–DNA interactions, identifying TF binding sites, and characterizing histone marks that modulate gene expression epigenetically. While bioinformatic tools such as HOMER [9] and ROSE [3,10] facilitate the identification of SEs from ChIP-Seq data, enhancers are identified using other approaches [11]. Several databases, including SEdb 2.0 [12], SEanalysis [13], CenhANCER [14], ENdb [15], and EnhancerAtlas 2.0 [16], utilize ChIP-Seq data to identify SEs and provide information on enhancer–target gene associations, diseases, TFs, and cancer types. Given the close relationship between gene regulation and chromatin conformation, other assays like Hi-C, ChIA-PET, and HiChIP are also utilized to identify enhancers. These methods elucidate chromatin structure, aiding in the validation of enhancer–target gene maps [17,18].

Non-coding RNA molecules, such as microRNA (miRNA), participate in gene regulation by silencing RNA and controlling post-transcriptional expression [19]. These approximately 21-nucleotide-long miRNAs target messenger RNA (mRNA) transcripts, regulating gene translation across various critical pathways. Experimental identification of miRNA genes and their targets has been challenging and involves laborious techniques leading to the development of computational algorithms [20] such as base-pairing patterns and thermodynamic stability of miRNA–mRNA hybrids, etc. [21]. Computational algorithms like TargetScan [22], TargetScanS [23], PicTar [24], miRanda [25], etc., along with databases like miRBase Targets [26], and TarBase5.0 [27], etc., aid in miRNA gene identification, target prediction, and integrating functional annotations. These databases are either based on literature mining or computational algorithms that use the aforementioned target prediction methods. However, depending on the algorithms used, computational methods incur varying amounts of false-positive results in identifying target mRNAs. Thus, the overlap of information across different databases is often low [21], necessitating experimental validation. Some databases like GOMir [28], miRecords [29], etc., integrate information from multiple prediction algorithms to achieve more accurate results, but this often worsens inference [30]. This necessitates experimental validation to check the interaction between miRNA and its target mRNA directly [21] or via computational methods that would learn from the true-positive results of existing large databases [31]. Recent databases like mirDIP 4.1 [32] claim improved accuracy by integrating predictions from multiple resources. Other methods such as MiRror [33] and miRGate [34] also provide integrated target predictions. The interaction between miRNAs and TFs involving auto-regulatory feedback loops is a common regulatory mechanism for controlling gene expression [35]. Experimental evidence supports the joint regulation of target genes by miRNAs and TFs, highlighting their role in gene expression control [36,37]. Many computational methods and databases identify networks to predict miRNA–TF, miRNA–mRNA, miRNA–TF–mRNA [38], and miRNA–disease regulatory relations [39]. Some of these methods are based on the networks that consider feed-forward, feed-backward, etc., loops that controls gene regulation [40].

Furthermore, SEs have emerged as key players in gene regulation, with experimental studies linking them to miRNA production, miRNA-associated diseases, and TFs [41,42,43] in addition to their pivotal roles in cell identity and disease progression. Although enhancers and SEs were experimentally found to be associated with disease genes, they have not been extensively computationally explored to identify their connections in gene regulation networks. Some databases are constructed based on the overlaps between SEs and (a) the neighborhood of promoter regions of target genes, (b) TF binding sites or genes encoding TFs, and (c) the neighborhood of the transcription start site of miRNAs [44,45]. SEs are typically identified using the ROSE or HOMER algorithms from different publicly available ChIP-Seq data [46] and integrated with other information based on the above approaches. Although these databases provide valuable information, they do not fully explore the combined impact of these regulators on gene expression. However, robust computational methods could be developed leveraging insights from multiple databases, to enhance our understanding of gene regulatory networks. These methods are expected to validate known findings from databases or experimental studies.

In this review, we first summarize our current understanding of computational methods unraveling gene regulation networks in connection to miRNAs and TFs obtained from ChIP-based experiments. We also outline the necessity of incorporating SEs, another source of ChIP-Seq-derived information, into the network and formulate open problems that highlight computationally motivating research areas. We broadly divide the research areas into three categories: (1) unraveling the indirect network of gene regulation, (2) identification of network motifs to increase the precision of target genes, and (3) enriched pathway identification by jointly dissecting gene regulators. This review is driven by the potential role of SEs in gene regulation networks, which is relatively overlooked compared to more established gene regulators such as TFs and miRNAs. Although experimental studies and databases have identified associations of SEs in gene expression networks, these methods are either very specific (as in experimental studies) or too generalized (as identified in databases), often measured across multiple studies/tissues/cell lines. Appropriate computational methods for the systematic identification of regulators in gene regulatory networks are crucial. Existing context-specific databases (summarized in Table 1) might be leveraged for developing such methods, and validation of the findings through experimental studies would be ideal. Moreover, validation from existing publications also provides evidence of the efficacy of the developed methods.

In the next sections, we provide three computationally motivating and challenging areas of research related to unraveling some of the well-known players (e.g., miRNAs and TFs) of gene regulatory networks, along with some recently popular ones like SEs (in Figure 1). With the existing methods available in each area (summarized in Table 2), we illustrate the potential insights and challenges that would be gained by incorporating SEs and provide motivational open problems for understanding gene regulation networks better. The emerging experimental studies illustrating gene regulation in the context of SE association (summarized in Table 3) provide a compelling reason to dissect SEs via computational methods to gain an improved understanding of gene regulation.

## 2. Unraveling Indirect Network of Gene Regulation

### 2.1. Advances in Computational Prediction of miRNA-Mediated Gene Regulation

Identifying the mechanisms orchestrating gene regulation is quite complex as they involve an intricate network of multiple cellular functions. Although experimental validations are required to confirm any cellular function, computational methods help in identifying the major players. To understand the role of miRNA in gene regulation, many challenges are encountered, such as the identification of miRNAs, prediction of target mRNA for the corresponding miRNA directly or indirectly via TFs or SEs, etc. Computational approaches for identifying miRNAs and predicting direct targets (with physical binding sites) of miRNAs have been extensively explored. However, the small overlap of miRNA targets across databases highlights the need for greater accuracy in miRNA–mRNA mapping. To acheive this, it is imperative to include more information. Given that gene expression is modulated by TFs and/or SEs at the transcriptional level and by miRNAs at the post-transcriptional level, and that these regulators can influence each other, dissecting their interplay might enhance our understanding of gene regulation [47]. Computational methods for identifying miRNA–TF–target gene interactions are not straightforward. Recently, Sayed et al. developed miRinGO [48] to predict gene ontology (GO) annotations indirectly targeted by miRNAs via TFs. Interestingly, this approach integrates (1) predicted TF targets of miRNAs using the TargetScan database [22] with (2) computationally predicted tissue-specific TFs associated with genes [49] using the PANDA [50] algorithm, to find indirect mRNA/gene targets of the corresponding miRNAs. The method PUMA [51] also uses PANDA to identify regulatory networks between miRNAs and target genes. PUMA extracts information from TargetScan and miRanda databases [25] to construct the network. However, this method does not consider the simultaneous regulatory effect of TFs on target genes.

PANDA is a widely used method for assessing agreement between diverse data types and refining predictions iteratively by exploring co-regulatory effects between networks. Initially combining gene expression, protein–protein interaction, and sequence motif data, PANDA constructs genome-wide regulatory networks with effector and affected nodes, using Tanimoto similarity for edge weight normalization. It updates networks iteratively until convergence, facilitating a comprehensive exploration and refinement of predictions. PANDA is a flexible method capable of exploring diverse data types and could be updated to incorporate SEs to understand gene regulatory networks. However, PANDA is a heuristic method and contains caveats related to the convergence of iterations; it could be modified to incorporate SE information with some effort.

### 2.2. Exploring the Role of SEs in miRNA-Mediated Gene Regulation

Although computational approaches for combining TFs with miRNAs to identify target mRNAs are available (such as miRinGO), SEs have not been extensively explored in this context, even though experimental methods have implicated SEs in miRNA production and miRNA-associated diseases [41]. SEs, together with histone marks, influence tissue-specific, evolutionarily conserved atlases of miRNA expression and function. The role of SE constituents in boosting cell-specific miRNA production has been validated by CRISPR/Cas9 assays [41]. SEs have been implicated along with key oncogenes in multiple cancer types [8,10]. The discovery of cancer hallmarks obtained from dissecting cancer-related miRNAs associated with SE alterations has paved the way for identifying SE–miRNA biomarkers [41,52]. Although experimental methods implicate SEs in miRNA processing, SE–miRNA regulatory relations have not been extensively computationally explored. On the other hand, the SE–TF regulatory network, which plays a crucial role in the carcinogenesis of malignant tumors [53], has not been explored in connection with target gene prediction for miRNAs. Recently, a few databases identifying enhancers [45], and SEs [12] using ChIP-Seq data have become available, which might be explored to better understand the interplay between genes and their regulators. Thus, ChIP-Seq data hold great potential for dissecting the indirect regulatory role of miRNAs on target genes.

## 3. Identification of Network Motifs to Increase the Precision of Target Genes

### 3.1. Advances in Network Motifs for Enhanced Gene Target Precision

Transcription networks regulate cellular responses of living cells through different biochemical wiring patterns called network motifs. One such motif is the feed-forward loop (FFL). In a regulatory network of genes altered by an miRNA and a TF (or two miRNAs) via an FFL, a target gene is regulated by both the miRNA and the TF (or the miRNAs). In an FFL with miRNA–TF co-regulators, the TF often regulates the miRNA, is regulated by the miRNA, or both (see Figure 2). Computational methods have identified these FFLs and hubs of target genes, each potentially targeted by multiple miRNAs and TFs [54]. Based on statistical ranks of the computationally predicted FFLs, dChip-GemiNI [55] identifies differential gene and miRNA expression between two biological conditions such as normal and cancer. Other methods [39,56] also explore the crosstalk between these two regulators and their targets. Moreover, to increase the precision of gene regulatory networks, it is important to understand the directionality of FFLs. These can either speed up the response time of the target gene expression following stimuli in one direction but not in the other. Such FFLs are called incoherent FFLs (see Figure 2A). Another type of FFL, called coherent FFL (see Figure 2B), acts as sign-sensitive delays. Since an FFL has three nodes, each edge can have two kinds of interactions (activator or repressor), resulting in eight different structural configurations of FFLs. Identification of the FFL structure [57] might provide insights into the architecture of the gene regulatory network.

### 3.2. Unresolved Issues in Modeling SEs and Enhancers within Network Motifs

On the other hand, enhancers are regulatory elements that regulate the expression of protein-coding genes or miRNAs by recruiting TFs in a tissue-specific manner. To activate adjacent genes, enhancers tend to loop to them [58]. However, many enhancers map to target gene promoters located large distances away by forming loops in the three-dimensional nuclear space to achieve close spatial proximity to distant genes [59]. Given that promoter interactomes are highly cell-type-specific, enhancers have been linked with the active promoters of genes [60]. Additionally, SEs, which are often associated with high signals of active histone marks from ChIP-Seq experiments, play an important role in the miRNA production. In cancer cells, miRNAs with SE gain are found to be more associated with oncogenic roles, while those with SE loss are correlated with tumor-suppressive behavior [41]. Experimental studies [3] have identified large enhancer domains densely occupied by TFs and Mediators, which are associated with genes encoding key regulators of cell identity. In pluripotent embryonic stem cells (ESCs), these domains, based on their sizes, TF density and content, ability to activate transcription, and sensitivity to perturbation, have been identified as SEs. In ESCs, alterations in TFs or Mediators cause preferential loss of expression of SE-associated genes compared to other genes. In more differentiated cells, SEs containing cell-type-specific master TFs are also associated with genes that define cell identity. It has been observed that genes encoding ESC master TFs are themselves driven by SEs, forming feedback loops where key TFs regulate their own expression [3]. For example, IKAROS, prominently associated with leukemia, collaborates with TFs and SEs via FFLs, triggering an aberrant gene expression program in a B-cell epithelial transition [61]. Experimental studies conducted over a decade by a scientific group identified the involvement of deregulated miRNAs, SEs, and TFs in natural killer/T-cell lymphoma (NKTL) pathways [62,63]. This provides a clear motivation for involving SEs or enhancers in identifying network motifs to predict target genes with increased precision. However, computational methods for identifying FFLs or other network motifs involving SEs, TFs, and miRNAs in transcription regulation have not been explored in detail.

Even though SEs have been implicated in influencing tissue-specific miRNAs, the exploration of network motifs in relation to transcription regulation has not been extensively studied. The database EnhFFL [44] provides FFLs and other network motifs in relation to SE and typical enhancer with TFs, miRNAs, and genes based on deterministic connections. However, networks might come with intrinsic and/or experimental uncertainties, necessitating an exploration of their stochastic properties. Such methods would account for the intrinsic uncertainties of the network building blocks. It is important to note that, functionally related network motifs are not identical to topologically related motifs. Topological networks identified by deterministic approaches might be contaminated with noise from experimental assays or databases. Stochastic networks can help circumvent the effect of such noisy observations through statistical or probabilistic modeling. Although some methods are available in conjunction with protein–protein interaction or gene regulation [64,65,66], methods dealing with miRNAs, enhancers or SEs, and TFs in connection to gene regulation have not been thoroughly explored.

## 4. Enriched Pathways Identification by Jointly Dissecting Gene Regulators

### 4.1. Advances in Pathway Enrichment through Integrated Gene Regulator Analysis

Gene regulation networks are complex phenomena that involve not only the regulation of a particular gene by its multiple regulators, such as TFs and miRNAs, but also the regulation of the expression of these regulators themselves. Computationally identifying associations between individual TFs or miRNAs and genes might not be sufficient, as the target gene may also be regulated by other TFs and/or miRNAs. On the other hand, any TF or miRNA often targets multiple genes. Studies have shown that these genes are co-expressed to varying degrees depending on the gene regulators [67]. For example, compared to miRNAs, genes targeted by the same TFs tend to be more co-regulated at both the mRNA and protein levels. However, genes sharing common TFs or miRNAs have been associated with the same disease. The complexity of gene regulation networks increases because some genes also regulate the expression of multiple miRNAs and TFs. Experimental studies provide important evidence confirming the role of the proto-oncogene c-Myc in regulating the expression of the TF E2F1 by altering the expression of a cluster of miRNAs [68]. Conversely, E2F1 regulates c-Myc, thus revealing a putative positive feedback circuit [69]. Both c-Myc and E2F1 regulate cell growth and death by inducing transcription and modulating signal transduction [69]. It has been found that miRNAs regulate, and are regulated by, E2F1, creating an auto-regulatory loop [70]. Collectively, these experimental studies reflect the complex interplay of various gene regulators in important pathways. Other experimental studies have confirmed the promotion of oncogenic MYC expression via SEs in the Wnt-signaling pathway [71]. Additionally, SEs associated with TFs and other complexes play an important role in gene regulation [61,63]. Thus, it is imperative to develop computational methods to systematically dissect information on gene regulators along with target genes to understand context-specific gene regulation. This might lead to the identification of functionally enriched pathways, which could, in turn, encourage experimental studies to explore in detail specific players of gene regulation identified in a given context. While some methods [72] leverage miRNA–TF co-regulatory networks to identify pathways under miRNA control and significantly enrich the proportion of true miRNA–target interactions, others have constructed tools for identifying enrichment analysis for miRNAs [73,74]. The availability of such methods for miRNAs underscores the importance of developing similar methods that includes SEs.

### 4.2. Challenges in Computational Identification of SE-Driven Pathways and miRNA Interactions

Apart from playing a prominent role in miRNA biogenesis [41], SEs have been implicated in altering the expression of genes key to cell identity in both normal and diseased cells. In particular, experimental studies [75] show that cancer cells more often acquire SEs at genes that foster tumorigenesis. In addition, these genes are sensitive to perturbations in oncogenic signaling pathways. Furthermore, the enhancers constituting an SE may individually respond to various signals, enabling the regulation of a single gene’s transcription by multiple signaling pathways. Multiple studies have unraveled the potential involvement of cancer-specific SEs in the dysregulation of signaling pathways [76]. Coupled with these observations, the role of SEs in driving the biogenesis of miRNAs crucial for cell identity via enhancement of both transcription and Drosha/DGCR8-mediated primary miRNA (pri-miRNA) processing [41] highlights the importance of identifying enriched pathways by jointly dissecting these gene regulators. Other studies also implicate miRNAs driven by SEs that are associated with diseases or pathways. For example, miRNAs driven by SEs positively regulate the Hippo pathway during liver development [77], the deletion of KLF6 SE inhibits cell proliferation in HepG2 cells via miR-1301 overexpression [43], and, in diseases such as black fever [78], miRNAs driven by SEs mediate immune suppression. Efforts have been made to relate SEs with miRNA-associated gene regulation in pan-cancer analyses. Contrary to the common inverse proportionality relation in expression between miRNAs and mRNAs, studies [79] showed that top-ranked positively correlated miRNA/gene sites are more likely to form SEs in major human cancers. Although such studies highlight the potential of decoding the relationship between SEs and associated miRNAs to understand gene regulation, computational methods to systematically explore such relationships are still lacking. A plethora of associations are identified by the SEanalysis 2.0 [13] database, which connects SEs, pathways, TFs, and genes. The existence of such a database provides important evidence for identifying the inter-relationships between gene regulators in functionally relevant pathways. Although computational methods for identifying pathways have been explored in the context of miRNAs and TFs [72], methods involving SEs are still unavailable.

**Table 1 ncrna-10-00045-t001:** Super-enhancer (SE)-related databases and softwares.

Database/Softwares	Feature
ROSE [3,10]	Pipeline identifying SEs from ChIP-Seq data; separates SEs and typical enhancers
HOMER (v5.1) [9]	Software for motif discovery and ChIP-Seq analysis; identifies enhancers and SEs
SEdb 2.0 [12]	Database for SE resource and annotate the potential roles in gene transcription
SEanalysis 2.0 [13]	Web server for identifying association connecting SEs, pathways, TFs, and genes
CenhANCER [14]	Database for cancer enhancers from primary tissues and cell lines
ENdb [15]	Manually curated database of experimentally supported enhancers
EnhancerAtlas 2.0 [16]	Database with enhancer annotation across nine species
TRmir [46]	Database for miRNA-related transcriptional regulation, especially typical enhancer and SE
EnhFFL [44]	Database for enhancer-related FFLs based on deterministic connections
EnhancerDB [45]	Database for enhancer-related transcriptional regulatory associations

**Table 2 ncrna-10-00045-t002:** Computational methods related to miRNA, mRNA, TF, and sequence motif to understand gene regulation network.

Methods	Feature
miRinGO [48]	Accumulate information from databases on TFs associated target genes and miRNAs; then combine them to predict genes that miRNAs target via TFs
PANDA [50]	A message-passing model integrating protein–protein interaction, gene expression, and sequence motif data to predict regulatory relationships
PUMA [51]	Identify gene regulatory networks under miRNA control using PANDA and target genes
Sonawane et al. [49]	Computationally predict tissue-specific TF associated with genes using PANDA
dChip-GemiNI [55]	Statistically ranks computationally predicted FFLs to account for differential gene and miRNA expression between two biological conditions
FFLtool [56]	A web based tool for detecting FFL of TF–miRNA–target regulation in human
Mangan and Alon [57]	Theoretically analyze the functions of all possible structural types of FFLs
Jiang et al. [64]	Identify network motif using stochastic networks
Yeger-Lotem et al. [65]	Developed algorithms for detecting networks motifs with two or more types of interactions
Kashtan et al. [66]	Algorithms for detecting network motif generalizations
Prompsy et al. [72]	Leveraged miRNA–TF co-regulatory networks to identify pathways under miRNA control, and significantly enriched the proportion of true miRNA–target interactions
MiEAA [73]	A web-based application for miRNA set enrichment analysis and annotation
miRFA [74]	Pipeline for biomarker discovery involving mature miRNAs
Shalgi et al. [54]	Identifies miRNA–TF regulatory network

**Table 3 ncrna-10-00045-t003:** Experimental studies illustrating gene regulation in context of SE association.

Studies	Feature
Whyte et al. [3]	SEs play key roles in the control of mammalian cell identity; formation of SE-driven feedback loops; regulation of SE-associated gene expression via master TFs
Hnisz et al. [75]	SEs are occupied more frequently by terminal TFs of the Wnt-, TGF-b-, and LIF-signaling pathways in ESCs/cancer cells; and SE-driven genes respond to manipulation of these pathways compared to typical enhancers
Hnisz et al. [8]	Cancer cells generate SE at oncogenes and other genes related to tumor pathogenesis
Lovén et al. [10]	SEs are associated with critical oncogenic drivers in cancer cells
Suzuki et al. [41,52]	SEs potentially drive the biogenesis of miRNAs crucial for cell identity via enhancement of both transcription and Drosha/DGCR8-mediated primary miRNA processing
Ri et al. [43]	Over-expression of miR-1301 induced by deletion of KLF6 SE inhibits cell proliferation in HepG2 cells
Liang et al. [53]	SE–TF regulatory network plays a crucial role in the carcinogenesis of malignant tumor
Javierre et al. [60]	Promoter interactions are highly cell-type-specific and enriched for association between active promoters and epigenetically marked enhancers
Hu et al. [61]	IKAROS, prominently associated with leukemia, collaborates with TFs and SEs via FFL, and triggers aberrant gene expression program in a B-cell epithelial transition
Zhou et al. [63]	SE-driven TF gene mediates oncogenesis in Natural Killer/T Cell Lymphoma
Scholz et al. [71]	WNT signaling activates MYC expression via SE in cancer cells
Zhang et al. [77]	miRNAs driven by SEs positively regulate Hippo pathway during liver development
Das et al. [78]	miRNAs driven by SEs mediate immune-suppression
Tan et al. [79]	miRNAs/genes with positive correlations tend to form super-enhancer-like regions
Turunen et al. [80]	Synergistic role of miRNAs and TFs coinciding with SEs are associated with Hippo signaling pathway

## 5. Discussion

miRNAs and TFs have been extensively studied in connection with gene regulation. Aberrations in these gene regulators have been implicated in various diseases. In this review, we highlighted the role of SEs/enhancers, which potentially provide much more information, in addition to TFs, to unravel gene regulatory networks under miRNA control. The rationale behind this review was to address the lack of systematic statistical/computational approaches in three primary domains pertinent to uncovering gene regulatory networks, which stand to benefit from the valuable information harbored in SEs/enhancers. Recent experimental studies linking SEs to miRNA production and function, alongside their established role in regulating key genes, have motivated us to explore the potential of SEs/enhancers in bridging the current gap in our understanding of gene regulatory networks. Furthermore, the existence of dedicated databases for each of these distinct domains underscores the significance of the correlation between SEs/enhancers and other gene regulators.

Recent experimental studies have illustrated the synergistic role of miRNAs and TFs, in conjunction with SEs, in gene regulatory networks, particularly in cancer [80]. SEs are not only observed near genes with cell-type-specific functions but are also considered sensitive to alterations in chromatin-based mechanisms of gene regulation [8]. For example, in Burkitt’s lymphoma, strong enhancers come in close proximity to oncogenes via genomic rearrangements such as translocation. Additionally, compared to normal enhancers, SEs are enriched in sequence motifs corresponding to cell-type-specific master TFs. SEs have been implicated in driving the biogenesis of miRNAs that are crucial for cell identity through the enhancement of both transcription and Drosha/DGCR8-mediated pri-miRNA processing. CRISPR/Cas9 genomics revealed that SEs facilitate Drosha/DGCR8 recruitment and pri-miRNA processing, which boosts cell-specific miRNA production [41]. Tissue-specific and evolutionarily conserved atlases of miRNA expression and function are observed to be largely shaped by SEs and broad H3K4me3 domains identified by ChIP-Seq. Both well-studied cancer-related miRNAs [81] and other miRNAs with SE alterations have been associated with cancer hallmarks. Moreover, targeting SE components, such as disrupting SE structure or inhibiting SE cofactors, has attracted therapeutic interest across various cancers [76].

It is well known that a single miRNA can target multiple genes and mediate their post-translational regulation. However, accurately identifying these target genes experimentally and computationally remains a challenge. Numerous databases are constructed based on curated experimental gene targets and/or computationally predicted targets. The minimal overlap between these databases indicates the presence of false-positive predictions. The reason behind this discrepancy might be the lack of appropriate methods available to combine more players involved in the regulatory network. With the advent of NGS technology, we have access to a vast collection of high-throughput data, but understanding the complexity of the gene regulation network remains challenging. Computationally, TFs and miRNAs are extensively studied in relation to gene expression regulation due to their better biological understanding. However, SEs have not been explored as thoroughly as TFs and miRNAs, despite experimental validation of their involvement in gene regulatory networks. Although SEs could be identified from ChIP-Seq experiments similar to TFs, the identification of SEs is not straightforward [82]. Oftentimes, putative enhancers in close genomic proximity are termed SEs. Although the definition of SEs is not very clear, a few databases such as HOMER [9], ROSE [3,10], and EnhancerDB [45] are now available, which provide information on SEs and enhancers.

Here, we aimed to provide an overview of existing computational methods that uncover gene regulatory networks in connection with SEs, TFs, and miRNAs jointly. We also highlighted gaps in knowledge in three different areas as open problems for the scientific community to address. In relation to gene expression regulation, we first explored networks that dissect the roles of miRNAs, TFs, and gene expressions using data from different databases through integrative computational approaches. Many of these methods are constructed based on iteratively updating information from one data source using different data sources. Other methods use multiple databases and integrate partial information to get an overall picture of predicted associations. However, none of these methods have explored SEs in this context, despite their experimental implication in gene expression aberrations. Secondly, various network motifs or biochemical wiring patterns have been studied relating to genes and their regulators, such as TFs and miRNAs. Computationally identifying these patterns might provide greater insight into the gene regulatory network. Motifs such as FFL and others aid in understanding the actual direction of the flow of information. However, identifying motifs in connection with SEs or enhancers has been largely overlooked. The presence of databases such as EnhFFL, which curate motifs related to enhancers/SEs, emphasizes the importance of such information to the scientific community. These FFLs are created based on physical overlaps between either enhancer–TF–miRNA, enhancer–TF–gene, enhancer–TF, or enhancer–miRNA–gene across several tissue/cell lines in humans and mice. Lastly, we discussed the importance of identifying pathways through enrichment analysis. Since gene regulatory networks are quite complex, with multiple genes being targets of one or more miRNAs and vice versa, it is imperative to focus collectively on genes with similar functions and their regulators. Databases such as SEanalysis 2.0 curated from multiple ChIP-Seq data sources, provide connections between SEs, pathways, TFs, and genes, highlighting the evidence for identifying inter-relationships between gene regulators in functionally relevant pathways. Although computational methods [72] for identifying pathways associated with miRNAs and TFs exist, similar approaches involving SEs are still needed to provide substantial insight into gene regulatory networks.

In conclusion, while significant progress has been made in understanding gene regulation networks involving TFs and miRNAs, integrating SEs into computational analyses remains an open research area. In this review, we hypothesized that developing computational methods to incorporate SEs into regulatory network predictions could enhance our understanding of gene regulation mechanisms and facilitate the identification of functionally relevant pathways in health and disease.

## Figures and Tables

**Figure 1 ncrna-10-00045-f001:**
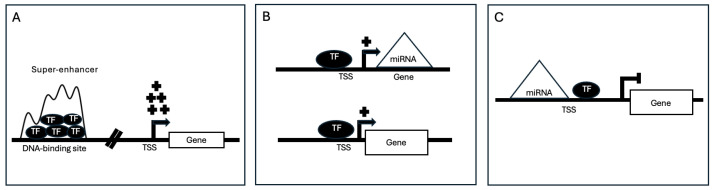
Graphical representation of gene regulation in the context of super-enhancer (SE), transcription factor (TF), miRNA genes, and target genes. (**A**) SE located away (denoted by breaks on the DNA) from the transcription start site (TSS) of target genes highly regulates (here, activates) the expression of the corresponding genes via TFs and other protein complexes; (**B**) TFs bind to promoter regions of miRNA and other genes to regulate (here, activate) the expression of the corresponding genes; (**C**) miRNA and TF, together, regulate (here, repress) gene expression. The number of + symbols represents the intensity of transcriptional activity, with a higher count indicating increased activity.

**Figure 2 ncrna-10-00045-f002:**
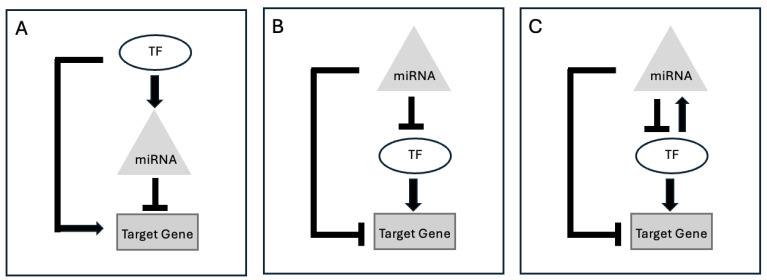
Graphical representation of gene regulation via miRNA–TF FFLs; (**A**) TF-mediated FFL: TF simultaneously activates miRNA and target gene but miRNA represses target gene, (**B**) miRNA-mediated FFL: miRNA represses TF and target gene but TF activates target gene, and (**C**) composite FFL: miRNA represses TF and target gene but TF activates both miRNA and target gene.

## Data Availability

No new data were created or analyzed in this study. Data sharing is not applicable to this article.

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
