# Peer review of "Predicting the Effect of miRNA on Gene Regulation to Foster Translational Multi-Omics Research—A Review on the Role of Super-Enhancers"

_ncrna, 2024, doi:10.3390/ncrna10040045_

Round 1
Reviewer 1 Report
Comments and Suggestions for Authors
Review of the manuscript titled, “Predicting the effect of miRNA on gene regulation to foster translational multi-omics research - a review on the role of Super-Enhancers” by Sarmistha Das and colleagues. The author reviewed the potential role of Super-enhancer (SE) in miRNA-related gene regulatory network. The aim and the plan of the study are innovative. However, a couple of concerns need to be fully addressed before being published.
specific comments
1、line21-41: This part is completely consistent with the abstract, and the author should rewrite or delete it.
2、The title of 2.1, 3.1, and 4.1 is completely consistent, and the same problem also occurs in 2.2, 3.2, and 4.2. The author should provide a more specific current situation and dilemma as the title.
3、The author should cite more papers from the past last years(2022-2024), and the description of current progress is not detailed enough.
Author Response
The authors express their gratitude to the reviewers for highlighting areas for improvement in our review paper. We value the constructive suggestions offered by the reviewers aimed at enhancing the manuscript's organization, readability, and overall clarity. In this response, we have carefully considered and addressed the suggestions and comments and have highlighted the changes made in the revised manuscript.
Comments 1: line21-41: This part is completely consistent with the abstract, and the author should rewrite or delete it.
Response 1: The authors thank the reviewer for pointing this out. Although the authors are confused about this comment, we found that the abstract got repeated in the lines 21-41. We have removed these lines.
Comments 2: The title of 2.1, 3.1, and 4.1 is completely consistent, and the same problem also occurs in 2.2, 3.2, and 4.2. The author should provide a more specific current situation and dilemma as the title.
Response 2: The authors thank the reviewer for pointing this out. Although the authors are confused about this comment, we have tried to provide more explanatory sub-section titles for 2.1, 3.1, 4.1, 2.2, 3.2, and 4.2 that highlights the content of the corresponding sub-sections. We renamed the subsections as follows:
- The Subsection 2.1 heading is changed from “Status” to “Advances in computational prediction of miRNA-mediated gene regulation”
- The Subsection 2.2 heading is changed from “Open problem” to “Exploring the role of SEs in miRNA-mediated gene regulation”
- The Subsection 3.1 heading is changed from “Status” to “Advances in network motifs for enhanced gene target precision”
- The Subsection 3.2 heading is changed from “Open problem” to “Unresolved issues in modeling SEs and enhancers within network motifs”
- The Subsection 4.1 heading is changed from “Status” to “Advances in pathway enrichment through integrated gene regulator analysis”
- The Subsection 4.2 heading is changed from “Open problem” to “Challenges in computational identification of SE-driven pathways and miRNA interactions”
Comments 3: The author should cite more papers from the past last years(2022-2024), and the description of current progress is not detailed enough.
Response 3: We agree that we have not provided any reference paper from 2024. But we already have cited 7 papers from 2022-2023. Since this area has huge publication, we have provided reference to some of the major works only. These references covered all the sections from 1-4 in the manuscript. However, the authors could not find any computational or experimental (biological) publication from the year 2024 illustrating a major addition to the focus of interest of this review.
Reviewer 2 Report
Comments and Suggestions for Authors
The manuscript titled "Predicting the effect of miRNA on gene regulation to foster translational multi-omics research - a review on the role of Super-Enhancers" discusses the need for the inclusion of SEs in the evaluation of miRNA/TE regulatory networks.
The topic is interesting and important, as addressing this question would indeed reveal molecular regulatory connections that remained hidden so far. The review provides a detailed description on the currently available tools and methods for the analysis of the miRNA/TE/SE regulatory networks, highlighting their failings and shortcomings. The authors also discuss the evidence that underlines the need for the incorporation of SEs in these networks. These will be useful to initiate further studies in this direction, but I feel that the manuscript could be improved at some areas.
1. The text is redundant at several places, repeating the same information or notion without significant new information, making reading somewhat cumbersome. For example, the last two paragraphs of the Introduction (starting from line 117) contain almost all important information on the following sections in such a condensed manner that it hinders understanding of the message.
Instead of these repetitions, it would be better to provide a more detailed mechanistic review on how miRNAs, TEs and SEs can interact in gene expression regulation. This could involve upgrading Figure 1. to provide a more nuanced visualisation of this (maybe through multiple panels).
2. Alltogether, the manuscript would benefit from the addition of more explanatory figures, e.g. about the network motifs.
3. Extensive English editing is needed, as the text contains several grammatical errors, typically with the matching of singular and plural cases.
Comments on the Quality of English LanguageExtensive English editing is needed, as the text contains several grammatical errors, typically with the matching of singular and plural cases.
Author Response
The authors express their gratitude to the reviewers for highlighting areas for improvement in our review paper. We value the constructive suggestions offered by the reviewers aimed at enhancing the manuscript's organization, readability, and overall clarity. In this response, we have carefully considered and addressed the suggestions and comments and have highlighted the changes made in the revised manuscript.
Comments 1: The text is redundant at several places, repeating the same information or notion without significant new information, making reading somewhat cumbersome. For example, the last two paragraphs of the Introduction (starting from line 117) contain almost all important information on the following sections in such a condensed manner that it hinders understanding of the message.
Instead of these repetitions, it would be better to provide a more detailed mechanistic review on how miRNAs, TEs and SEs can interact in gene expression regulation. This could involve upgrading Figure 1. to provide a more nuanced visualisation of this (maybe through multiple panels).
Response 1: Thank you for pointing this out. We agree with this comment.
To avoid repetition, we have removed the paragraph (lines 21-41 in the old manuscript) before Introduction section. It contained summary of this review paper.
To remove redundancy, we have re-written the last two paragraphs of the Introduction section.
Lines 124-138 (in the original manuscript) are removed. These lines described overall idea of the mentioned three broad areas of existing research.
To maintain the flow of this paragraph, we have re-written line 139-140 as “This review is driven by the potential role of super-enhancers (SEs) in gene regulation networks that is relatively overlooked compared to more established gene regulators such as transcription factors (TFs), microRNAs (miRNAs).”
Figure1, is updated to include different panels as suggested by the reviewer to depict the interaction of miRNAs, TFs, and SEs with gene expression regulation.
Comments 2: Alltogether, the manuscript would benefit from the addition of more explanatory figures, e.g. about the network motifs.
Response 2: Thank you for pointing this out. We agree with this comment. We have added Figure2 to explain the network motif.
Comments 3: Extensive English editing is needed, as the text contains several grammatical errors, typically with the matching of singular and plural cases.
Response 3: Thank you for pointing this out. We have carefully corrected the grammatical errors. The errors are highlighted in the revised manuscript.
Reviewer 3 Report
Comments and Suggestions for Authors
The manuscript is very well written and comprehensive
I only have minor comments
Alot of typo mistakes and grammatical mistakes have to be corrected throughout the manuscript
The use of abbreviations should be re-considered and to be eplained at their 1st appearance only
The methodology is sound, and the results are clearly presented and discussed. However, the conclusion section could be strengthened to provide a more impactful and comprehensive closure to the work.
Otherwise, the manuscript is worth the publication in ncRNA
Comments on the Quality of English LanguageMinor edits
Author Response
The authors express their gratitude to the reviewers for highlighting areas for improvement in our review paper. We value the constructive suggestions offered by the reviewers aimed at enhancing the manuscript's organization, readability, and overall clarity. In this response, we have carefully considered and addressed the suggestions and comments and have highlighted the changes made in the revised manuscript.
Comments 1: Alot of typo mistakes and grammatical mistakes have to be corrected throughout the manuscript
Response 1: Thank you for pointing this out. We have carefully corrected the grammatical errors. The errors are highlighted in the revised manuscript.
Comments 2: The use of abbreviations should be re-considered and to be eplained at their 1st appearance only
Response 2: Thank you for pointing this out. We have carefully provided the abbreviation at their first appearance only as per reviewer’s suggestion.
Comments 3: The methodology is sound, and the results are clearly presented and discussed. However, the conclusion section could be strengthened to provide a more impactful and comprehensive closure to the work.
Response 3: Thank you for pointing this out. As per reviewer’s suggestion to strengthen the conclusion, we have rewritten the concluding line of the manuscript. We have added the following sentence to conclude the manuscript. “In this review, we hypothesize that developing computational methods to incorporate SEs into regulatory network predictions could enhance our understanding of gene regulation mechanisms and facilitate the identification of functionally relevant pathways in health and disease.” We also removed the previous concluding sentence, “Developing computational methods to incorporate SEs into regulatory network predictions might enhance our understanding of gene regulation mechanisms and facilitate the identification of functionally relevant pathways in health and disease.”
Round 2
Reviewer 1 Report
Comments and Suggestions for Authors
I recommend the manuscript for publication.
Reviewer 2 Report
Comments and Suggestions for Authors
The authors responded to my comments and updated the manuscript. The current version has a much better consistency and readability, making it acceptable for publication.